# Expanding Possibilities for Foreign Gene Expression by Cucumber Green Mottle Mosaic Virus Genome-Based Bipartite Vector System

**DOI:** 10.3390/plants13101414

**Published:** 2024-05-19

**Authors:** Anirudha Chattopadhyay, A. Abdul Kader Jailani, Anirban Roy, Sunil Kumar Mukherjee, Bikash Mandal

**Affiliations:** 1Advanced Centre for Plant Virology, Division of Plant Pathology, Indian Agricultural Research Institute, New Delhi 110012, India; anirudhbhu@sdau.edu.in (A.C.); anirbanroy75@yahoo.com (A.R.); sunilmukherjeeudsc@gmail.com (S.K.M.); 2Pulses Research Station, Sardarkrushinagar Dantiwada Agricultural University, Sardarkrushinagar 385506, Gujarat, India; 3Plant Pathology Department, North Florida Research and Education Center, University of Florida, Quincy, FL 32351, USA; 4Plant Molecular Biology Group, International Centre for Genetic Engineering and Biotechnology, New Delhi 110067, India

**Keywords:** CGMMV, deconstructed genomes, trans-replication, movement, expression, virus-based vector, *N. benthamiana*

## Abstract

Expanding possibilities for foreign gene expression in cucurbits, we present a novel approach utilising a bipartite vector system based on the cucumber green mottle mosaic virus (CGMMV) genome. Traditional full-length CGMMV vectors face limitations such as a restricted cargo capacity and unstable foreign gene expression. To address these challenges, we developed two ‘deconstructed’ CGMMV genomes, DG-1 and DG-2. DG-1 features a major internal deletion, resulting in the loss of crucial replicase enzyme domains, rendering it incapable of self-replication. However, a staggered infiltration of DG-1 in CGMMV-infected plants enabled successful replication and movement, facilitating gene-silencing experiments. Conversely, DG-2 was engineered to enhance replication rates and provide multiple cloning sites. Although it exhibited higher replication rates, DG-2 remained localised within infiltrated tissue, displaying trans-replication and restricted movement. Notably, DG-2 demonstrated utility in expressing GFP, with a peak expression observed between 6 and 10 days post-infiltration. Overall, our bipartite system represents a significant advancement in functional genomics, offering a robust tool for foreign gene expression in *Nicotiana benthamiana*.

## 1. Introduction

The infection process of plant RNA viruses is highly dynamic. These viruses employ various strategies to establish themselves within plant cells. This includes processes like frame-shifting, overlapping genome structures, and the production of subgenomic RNAs (sgRNAs). Additionally, during pathogenesis, many plant RNA viruses generate defective RNAs (D-RNAs) and satellite RNAs (satRNAs) to modulate the infection process. These D-RNAs and satRNAs are formed through a recombination and rearrangement of the wild-type virus genome during faulty replication, rendering them incapable of self-replication [1,2]. Instead, they rely on the parental wild-type virus acting as a helper virus for replication, when both are present in the same cell. These defective RNAs typically retain the necessary cis-regulatory elements for their replication [3] and encapsidation [4], which may facilitate their local or systemic movement within the plant as progeny particles.

Even though these defective RNAs represent only a portion of the viral genome, understanding their functionality is crucial for developing gene-delivery vehicles. Such vehicles can be used for expressing specific genes of interest, either for functional genomics studies or therapeutic purposes. For the expression of foreign genes, two main strategies are commonly employed: the “full” virus vector strategy and the “deconstructed” virus strategy [5]. In the full virus genome-based strategy, the gene of interest is inserted within the coding frame of the full virus genome, allowing foreign genes to be expressed along with the functional virus as fused mRNA or protein. However, this approach often faces challenges such as low stability, limited expression levels, and restricted cargo capacity, making it less suitable for industrial-scale applications [6]. To overcome these limitations, deconstructed virus genome-based vectors have gained popularity. In this approach, the virus genome is redesigned to retain only the essential regulatory elements for replication and translation, while eliminating other gene elements necessary for various viral functions such as symptoms, transmission, and movement. Consequently, deconstructed virus vectors rely on a “parental” helper virus for genome functions like replication and translation. This strategy offers a stable and efficient expression of large and multiple foreign proteins, is easy to engineer, and does not pose biosafety concerns associated with the loss of virus function. Several deconstructed virus vectors have been developed using the genomes of popular viruses, with each designed for specific host systems.

In this context, the cucumber green mottle mosaic virus (CGMMV), a member of the Tobamovirus genus with a single-stranded, monopartite RNA genome, is an attractive candidate. CGMMV infects certain cucurbits and *Nicotiana benthamiana*, a model plant, causing mild symptoms in a large number of hosts [7]. Therefore, it holds potential as a biological toolkit for cucurbitaceous crops. The CGMMV genome consists of a single-stranded, positive-sense RNA (approximately 6.4 kb) with four overlapping open reading frames (ORFs) encoding four proteins: two replication proteins (129 kDa and 186 kDa), one 30 kDa movement protein (MP), and a 17–18 kDa coat protein (CP) [8]. The 5′ terminus of the CGMMV genome plays a crucial role in expressing replicase enzymes, while two 3′ co-terminal subgenomic RNAs (sgRNAs) express the MP and CP [9]. These sgRNAs are naturally generated due to erroneous replication mechanisms and are shorter than their cognate genomic RNAs, often including 5′ regulatory sequences. The production of sg mRNAs is a strategy employed by eukaryotic RNA viruses for the quantitative and temporal expression of proteins [10]. This strategy can also be valuable for the heterologous expression of other proteins, using a CGMMV-based expression vector.

However, limited efforts have been made to utilise CGMMV as an expression vector. Initial attempts focused on expressing smaller foreign genes (approximately 100 to 150 nt long) using the full-length CGMMV genome [11]. Later, Zheng et al. [12] attempted to express GFP (~800 nt) but got a weak expression. Subsequent efforts, like the one conducted in our laboratory, aimed to express larger genes such as GFP (approximately 800 nt). However, achieving a strong expression was challenging, especially when the foreign gene was inserted at the end of the CP gene. More promising results were obtained when the foreign gene was inserted at a specific location within the CP gene [13]. Nevertheless, instability in expression beyond 5–10 days post-inoculation (DPI) remained a challenge, highlighting the limitations of full-length viral genome-based vectors. Additionally, the lack of suitable restriction sites and undefined regulatory elements within the CGMMV genome, along with difficulties in loading multiple foreign genes, further restrict its utility.

To address these limitations, our goal is to design a new expression system using CGMMV. Our hypothesis is twofold: (1) deconstructing the CGMMV genome will lead to the development of shorter replicons that can replicate with the help of the main virus, and (2) these short replicons can serve as the basis for a new expression system capable of accommodating large and multiple foreign genes. However, before utilising them as potential expression vectors, it is essential to artificially create deconstructed genomic RNAs of CGMMV. These deconstructed RNAs (dRNAs) should retain minimal regulatory sequences, while preserving their biological properties, including in planta replication, movement, and persistence. This study will pave the way for a more versatile and efficient expression system using CGMMV.

## 2. Materials and Methodology

### 2.1. Designing of Deconstructed Virus Genomes and Analysing Their RNA Secondary Structures

Initially, we designed a deconstructed RNA genome (DG-1) from the CGMMV genome, based on the consensus sequence alignment of the previously reported dRNA genome structure of TMV, which included nucleotides 1–841 from the 5′ terminal and 5182–6424 from the 3′ terminal of the genome [14]. We used the Mfold Web server (http://mfold.rna.albany.edu/) to predict the RNA secondary structure of both the wild-type CGMMV and its deconstructed genome [15]. The input files included the complete RNA genome sequence and the dRNA genome of CGMMV. The secondary structure of dRNA was predicted at 37 °C using Mfold version 2.3 [15,16]. We selected potential stable structures with minimal Gibbs free energy from the output file and conducted a comparative analysis to predict RNA pseudoknots and the putative regulatory regions located at the 5′ and 3′ terminals of the genome. This analysis helped us to determine the possibility of trans-replication of the deconstructed genome.

Subsequently, we designed another deconstructed genome (DG-2) of CGMMV based on the in silico prediction of cis-regulatory elements within the CGMMV genome. Our in silico prediction identified the specific cis-regulatory elements crucial for replication and translation. These elements were primarily located within the 5′ and 3′ untranslated regions (UTRs) of the genome and the subgenomic promoter region of the MP and CP [17]. We merged these sequences to create a complete replicative deconstructed genome, DG-2. To facilitate the insertion of multiple foreign genes, we incorporated some restriction sites at various locations within the DG-2 genome. We used the Mfold web server (http://mfold.rna.albany.edu/) to predict the RNA secondary structure of DG-2 and compared it with that of the wild-type CGMMV. Similar to DG-1, we used the complete RNA genome sequence and the dRNA genome of CGMMV as input files, selecting potentially stable structures with minimal Gibbs free energy from the output file. This comparative analysis aimed to identify the conservation of regulatory elements located at the 5′ and 3′ terminals of the genome.

### 2.2. Development of Deconstructed Virus Genomes

The deconstructed plasmid (pDG1) was derived from the infectious clone (pBP4) of CGMMV-BgDel isolate (CGMMV-BP4) [13,18] by deleting the original plasmid construct (pBP4), using an LC-PCR-based cloning strategy [19,20]. This mechanism involved a one-step PCR, using two long primers (P5R and P6F) designed with a high GC content and sharing some degree of homology at their 5′ end (Appendix A). Another extra primer (P7) was used for self-circularisation. The PCR reactions were performed using a 50 µL master mix containing 1× PCR buffer, 0.8 mM dNTPs, 1 U hi-fidelity Phusion Taq DNA polymerase, 100 ng of the vector (plasmid), and 1 mM of each primer. The thermal cycling included 1 cycle at 98 °C for 5 min; 20 cycles at 98 °C for 20 s and at 65–70 °C for 8 min (@ 1 kb/30 s); and a final hold at 4 °C. After the PCR, the products were treated directly with the Dpn1 enzyme at a rate of 1U per 50 µL for 1 h 30 min without further purification. The resulting products were then transformed into highly efficient competent *E. coli* DH5α cells (10^8^ to 10^9^ cfu/µg). A colony PCR was used to screen for colonies containing the desired plasmid constructs, and positive colonies were further verified through restriction digestion of the plasmids. The PCR-derived deletion of the constructs was confirmed via sequencing, to ensure the presence of only the desired modifications.

The second deconstructed genome, DG-2, was designed by incorporating the partial genome sequence of CGMMV within the 2 × 35 s promoter and ribozyme (Rz) site with a Nos terminator. After synthesis, the entire cassette (approximately 4062 nt) was incorporated into the pUC57 vector backbone (2.7 Kb) using the KpnI and SacI restriction sites. Subsequently, the cassette was directly cloned into pGreen II (0029) using the same restriction sites. This process removed the entire multiple cloning sites (MCS) from the pGreenII (0029) backbone (4632 bp), allowing for the utilisation of these sites for further DG-2 genome modifications. Positive clones were verified through restriction digestion using the BamH1 and HindIII restriction enzymes, releasing desirable 2.78 kb fragments only from the vector backbone (4.7 Kb). The confirmed plasmid constructs were transferred into Agrobacterium GV-3101, along with the pSoup helper plasmid, via electroporation. The co-transformation of pGreen and pSoup was conducted at equal concentrations of both constructs, with only Kanamycin (not tetracycline) and Rifampicin for the selective screening of the transformed Agrobacterium colonies. The transformed Agrobacterium colonies were confirmed via a colony PCR, and positive colonies were maintained for further use.

### 2.3. Plant Inoculation of Deconstructed Virus Genomes along with Wild-Type Virus Strains

The cDNA plasmid constructs of the deletion mutant, deconstructed genome-1 (pDG1), and the synthetic genome construct (pDG2), along with the wild-type CGMMV infectious clones, viz., symptomatic constructs (pBP4) [20] as well as asymptomatic constructs (pBP7) were introduced into highly efficient competent cells of the Agrobacterium tumefaciens strain GV3101, using the freeze–thaw method [21]. Plasmid constructs (1–2 µg) were mixed with 200 µL of Agro-competent cells and incubated on ice for 20 min. The mixture was then subjected to freezing in liquid nitrogen for 20 s, followed by thawing at 37 °C for 5 min, with subsequent incubation on ice (0 °C) for 20 min. Afterwards, 5 mL of LB medium was added, and the culture was incubated at 28 °C for 3 h with continuous shaking (220 rpm). Transformed Agro cells were plated on appropriate antibiotic-containing plates (kanamycin and rifamycin) and incubated at 28 °C. The presence of recombinant clones in the Agro-culture was confirmed via a colony PCR, using gene-specific primers. The confirmed colonies were picked and individually grown in 5 mL of LB broth containing 50 μg/mL kanamycin and 15 μg/mL rifampicin at 28 °C for 18–24 h, with shaking at 200 rpm. The overnight Agro-culture was centrifuged at 5000 rpm for 10 min and resuspended in 50 mL of agroinfiltration buffer containing 10 mM 2-(N-morpholino) ethane sulfonic acid (MES, pH 5.7), 10 mM MgCl_2_, and 150 µM Acetosyringone, to reach a final OD of 0.6 for infiltration. The resuspended Agro-culture was further incubated at 28 °C for 1–3 h, with agitation at 50 rpm. Subsequently, the Agro-culture containing the pBP4 construct was infiltrated into 3-week-old *Nicotiana benthamiana* plants, using a sterile syringe on the abaxial leaf surface. To confirm the trans-replication of the deconstructed genomes, the Agro-cultures of the pDG1 (cDNA clone) and pDG2 (synthetic) constructs of CGMMV were either co-infiltrated (mixed at a 1:1 ratio) simultaneously with the wild-type CGMMV (pBP4/pBP7) or post-infiltrated (after 7 days post-inoculation of the wild-type CGMMV) into CGMMV-infected host plants. The infiltrated plants were maintained in a greenhouse with 16 h of daylight and 8 h of darkness at 25 °C, and observations were recorded. Each experiment was repeated three times for data replication and robustness.

### 2.4. Infectivity Assay of Wild-Type CGMMV

Leaf samples from pBP4-infected plants were harvested at 7 days post-inoculation (DPI) to confirm virus infection. The total plant RNA was isolated from symptomatic leaves, using a plant total RNA isolation kit (Promega, Madison, WI, USA) following the manufacturer’s protocol. For the cDNA synthesis, the total RNA was used as a template for a reverse transcriptase reaction with Superscript III (Invitrogen, Waltham, MA, USA), using a CGMMV-specific 3′ terminal primer (BM489R) according to the manufacturer’s protocol. The infectivity of the wild-type virus was detected through RT-PCR using CP gene-specific primers (BM-1178F and BM-1179R). After wild-type CGMMV infection, the deletion mutant was infiltrated in the case of post-infiltration.

### 2.5. Trans-Replication Assay of Deconstructed Genomes of CGMMV in Different Hosts

To detect the trans-replication of DG-1, RNA was extracted from infiltrated leaves and the systemic leaves of different host plants at different time intervals (7 DPI, 14 DPI), followed by a cDNA synthesis using the above-described protocol. The trans-replication of DG-1 was detected via RT-PCR, using virus-specific terminal primers (BM-486F and BM-489R). For additional confirmation, another duplex RT-PCR-based detection approach was employed, using virus-specific internal primer pairs (BM-1172F and BM-1173R), where the forward primer (BM-1172F) was designed from the flanking sequences (802–827 of DG-1 and BP4) located just prior to the junction/deleted site (835 nt) of DG-1, and the reverse primer (BM-1173R) was derived from the conserved sequence at 5981–6005 nt of BP4 and DG-1 (i.e., CP gene of CGMMV). On the other hand, the trans-replication and movement of DG-2 were detected using specific primers, BM-1265F and BM-1266R. RNA extracted from infiltrated and systemic tissues was used for testing. A simultaneous detection of helper CGMMV was carried out using primers 96F and 1268R. The PCR protocol was standardised, with denaturation at 94 °C for 3 min, followed by 35 cycles of denaturation at 94 °C for 30 s, annealing at 60 °C for 30 s, extension at 72 °C for 30 s, and a final extension at 72 °C for 10 min. Confirmation was made based on gel electrophoresis.

### 2.6. Movement Assay of Deconstructed Genomes of CGMMV

The in planta systemic movement of CGMMV’s dRNA was traced in different temporal and spatial scales. A total of ten CGMMV-infected plants were agroinfiltrated with DG-1, and leaf samples were harvested at various time intervals (from 1 DPI at 24 h intervals) and spatial scales (from infiltrated leaves to upper systemic leaves). RNA was extracted, and cDNA was prepared as per the manufacturer’s protocol. An RT-PCR was employed to detect DG-1 using terminal primers BM-486F and BM-489R. Systemic movement across different leaves was assessed at 14 DPI, to determine its distribution within the plant. The experiment was repeated three times for reliability. 

### 2.7. Vector Function Assay of Deconstructed Genomes within CGMMV-Infected *N. benthamiana*

#### 2.7.1. Gene-Silencing Assay of Deconstructed Genome-1-Expressing NbPDS Gene Sequence

##### Insertion of NbPDS Gene into DG-1 Construct, Agro-Transformation, and Plant Infiltration

A partial mRNA sequence (approximately 228 nt) of the phytoene desaturase (PDS) gene from *N. benthamiana* (NCBI accession number: EU165355) was cloned at the end of the CP region of the pDG1 construct in the sense orientation. To achieve this, a restriction- and ligation-free LC-PCR cloning technique was employed [20]. Long forward (P1F) and reverse primers (P2-R) were designed with sequences flanking both the CP and *NbPDS* gene (Appendix A). These primer pairs were used to amplify *NbPDS* gene segments in the first round of the PCR. The first round of the PCR followed a three-step protocol: 35 cycles of denaturation at 94 °C for 10 s, primer annealing at 52 °C for 30 s, and synthesis at 72 °C for 20 s, with a final extension at 72 °C for 10 min. The reaction mix included 5 µL of 5x reaction buffer, 2.0 µL of 2.5 mM dNTP mix, 1.0 µL of each forward and reverse primer (1.0 micromoles each), 1.0 U of Phusion Taq polymerase (NEB, Rowley, MA, USA), and DNase-free water to make up the volume to 25 µL. The primary PCR product was purified from the gel following the manufacturer’s protocol. In the second round of the PCR, the amplified *NbPDS* gene segment (−228 nt) served as long primers (P3) for insertion into the pDG1 construct (template) using a two-step LC-PCR protocol, along with an extra primer (P4) for the self-circularisation of the plasmid. This involved 20 cycles of denaturing at 94 °C for 10 s and primer annealing with a synthesis at 65 °C for 50 s per kb, followed by a final extension at 65 °C for 10 min.

After the completion of the PCR, the product was digested with the DpnI enzyme at 37 °C for 2 h, and then enzyme activity was stopped at 80 °C for 20 min. Five microlitres of DpnI-treated PCR product were directly transformed into *E.coli* JM109 chemically prepared competent cells, using a standard transformation protocol. The presence of the *NbPDS* gene at the end of the CP was further confirmed by using gene-specific primers (BM-1178F and BM-489R) and Sanger-based primer sequencing, using the CGMMV 3′ UTR primer (BM-489R) (Appendix A). Finally, the pDG(PDS)-1 construct was transferred into the Agrobacterium via electroporation, and the Agro-mobilised constructs were infiltrated into the CGMMV-infected *N. benthamiana* plants using the previously mentioned procedure.

##### Functional Validation of pDG(PDS)-1 Constructs

The phenotypic expression of the gene silencing was monitored during the 15–60 DPI period, starting from the appearance of photobleaching symptoms. The efficiency of the gene-silencing vector was estimated based on the percentage of plants showing photobleaching symptoms [22]. The infiltrated plants were tested for confirmation of the replication and movement of the pDG(PDS)-1 construct using an RT-PCR. A duplex RT-PCR was established using two pairs of primers: one set containing BM 1180R and BM 1269R for the specific detection of pDG(PDS)-1, while the other set, containing BM-96F and BM-1268R, was used for the specific detection of helper CGMMV simultaneously in the same PCR reaction.

##### Northern Blot Analysis for the Confirmation of the Deletion RNA Replicon

RNA extracted from different experimental samples was used for the Northern blot analysis. The study was conducted using the DIG-High Prime DNA Labelling and Detection Starter Kit II (Roche Applied Science, Penzberg, Germany). A digoxigenin-labelled positive-strand RNA genome-specific DNA probe was generated with DIG-High Prime and tagged with the 662 nt from the 3′ terminal of the CGMMV genome. This DIG-labelled probe was used for hybridisation with membrane-blotted RNA, which had been previously separated in an agarose gel using the standard methodology. The hybridised probes were immuno-detected with anti-digoxigenin-AP and Fab fragments, and then visualised with the chemiluminescence substrate CSPD, following the manufacturer’s protocol. The enzymatic dephosphorylation of the CSPD by alkaline phosphatase led to light emission at a maximum wavelength of 477 nm, which was recorded on X-ray films. Film exposure times ranged from 5 to 30 min. An in vitro RNA transcript was used as the positive control.

#### 2.7.2. In Planta Expression of Foreign Protein Using Deconstructed Genome-2

##### Detection of GFP Fluorescence through Confocal Microscopy

Initially, CGMMV-infected plants were selected for the agroinfiltration of the DG-2 construct. Starting from 2 DPI, plants were periodically analysed for GFP expression through confocal microscopy. Periodic observations of GFP expression were taken at 2-day intervals. pDG-2-infiltrated plants exhibited GFP expression in the inoculated area. CGMMV-infected plants, pDG-2-infiltrated plants, and PBS buffer-inoculated (mock) plants were chosen as control groups. GFP expression was confirmed by confocal microscopy, with a wavelength range of 490–510 nm.

##### Western Blot Assay for the Confirmation of GFP Expression

Protein extraction was performed using leaf tissue (0.8 g) from inoculated and systemic leaves of *N. benthamiana*. The tissue was initially ground to a fine powder in liquid nitrogen, using an autoclaved, pre-chilled mortar and pestle (−70 °C). The powder was then resuspended by inversion mixing in 300 µL of protein extraction buffer (0.1 M KH_2_PO_4_, pH 6.5, 0.5 mM PMSF, and 10 mM β-mercaptoethanol). Extracts were centrifuged for 20 min at 15,000 rpm at 4 °C, and the supernatant was collected and stored at −20 °C. Protein samples were prepared by mixing the crude protein with an equal volume of sample buffer and boiling it for 5 min in water. A Standard Sodium Dodecyl Sulphate Polyacrylamide Gel Electrophoresis (SDS-PAGE) was performed as per the protocol by Laemmli [23] to separate the proteins. The proteins were then transferred to a nitrocellulose membrane (NCM). The NCM was treated with a GFP-specific primary antibody, followed by a secondary antibody, goat anti-rabbit IgG conjugated with alkaline phosphatase (AP). After two washes with TTBS and one wash with TBS, the reaction was detected by adding the substrate (BCIP/NBT).

### 2.8. Quantitative Estimation of Replication Rate of Deconstructed Genomes

To quantify the replication level of the deletion mutants of CGMMV, a semi-quantitative experiment was conducted. DG-1 was agroinfiltrated into previously infected *N. benthamiana* plants. Ten plants were selected, based on their similar age and growth pattern. The 5th leaf of each infected plant was chosen for the agroinfiltration of DG-1. Small leaf discs (5 mm) were cut from the infiltrated part of all ten plants regularly at 24 h intervals, starting from the 1st DPI. These discs were combined for RNA extraction. The RNA was extracted following the previously described protocol, quantified, and treated with RNase-free on-column DNaseI to remove any influence from the infiltrated plasmid construct. The RNA samples were then used to synthesise first-strand cDNAs with an oligo(dT)20 primer using Superscripts III (Invitrogen, USA). An RT-PCR was performed to confirm the infectivity of CGMMV infectious clones in plants, using internal specific primers BM-1172F and BM-1173R. A quantitative RT-PCR was carried out to examine the RNA level of the DG-1 construct in the silenced, as well as healthy, plant leaves, using the primers BM 1180F and BM 1269R. The previously described PCR reaction mixture and program conditions were used.

Further, to quantify the DG-2 transcripts in inoculated plants, as well as control healthy transcripts, a qPCR was performed, targeting the GFP gene. The total RNA was extracted from 100 mg of plant tissue, using a plant RNA extraction kit (SV total RNA isolation system, Promega, USA). The qPCR was performed using specific primers to GFP (BM-1265F and BM-1266R) and actin1 (as an internal control) from *N. benthamiana* (BM-849F & BM-850R) with the SYBR green master mix from KAPA SYBR FAST qPCR Kits (KAPA Biosystem, Wilmington, MI, USA). The thermal cycling program for both their primers was 95 °C for 1 min, followed by 45 cycles at 95 °C for 20 s, 60 °C for 20 s, and 72 °C for 20 s. The reaction was performed in a qPCR machine Light Cycler 96 SW (Roche, Indianapolis, IN, USA).

## 3. Results

### 3.1. Design and Development of Deconstructed Genomes

We initiated the design and development of DG-1 (1988 nt) for CGMMV by using the previously reported dRNA of TMV as a reference. This dRNA from TMV, referred to as ∆HINC151 [24], retained 1–841 nt (equivalent to 258 amino acids) from the N-terminal, and 5182–6395 nt from the C-terminal, of its genome. Based on sequence alignment, we designed the dRNA of CGMMV to encompass only 1–835 nt of the N-terminal part and 5272 to 6424 nt of the C-terminal part (Figure 1). The internal sequence containing the helicase, RdRP, and movement protein was intentionally deleted, to eliminate critical functional properties such as self-replication and cell-to-cell movement.

To ensure its trans-replication with the assistance of wild-type CGMMV, we analysed its RNA secondary structure using the M-fold Web server. We identified a total of 35 folded RNA structures for DG-1 at 37 °C. Among them, we selected three structures with the lowest free energy value (ΔG = −562.10), all of which displayed conserved hairpin and pseudoknot-like structures at the 5′ and 3′ UTRs of the genome (Appendix A). Importantly, the deletion of the internal coding region did not affect the structural integrity of DG-1, which is essential for replication. The dot plot map of the structure also indicated the genome’s stability, based on minimum free energy. RNA viruses rely on stable secondary structures for their replication, transcription, or interaction with host factors. Perturbations in RNA secondary structure stability usually affect the replication fidelity of RNA viruses. Lower values of free energy (ΔG) indicate more stable secondary structures; thus, the deconstructed RNA molecules with lower ΔG values were less likely to undergo more structural instability. Interestingly, RNA structure prediction techniques based on free energy minimisation are typically used on a single RNA sequence [25]. Most energy-stable RNA secondary structure(s) of a plant virus genome contain canonical A:U, G:U, and G:C base pairs, to arrange the structure into a conventional helical form. Thus, it is considered that the lower the energy, the more stable is the structure [26].

For the generation of site-specific deletions within a particular DNA sequence, we employed the LC-PCR technique. This technique utilises primer sets with partially overlapping ends at a high annealing temperature (72 °C), to ensure precise site-specific binding at the target sequence, enabling accurate deletion and subsequent plasmid DNA construct ligation. This method eliminates the need for restriction digestion and ligation. The circularised PCR amplicons, which remained non-covalently closed, were transformed into *E. coli* competent cells after degrading the original plasmid molecules with DpnI enzyme treatment. A few colonies were obtained and confirmed using a colony PCR and plasmid-based PCR. These constructs were further validated through restriction digestion using BamHI and XbaI, with wild-type pBP4 plasmid serving as a positive control (Appendix A).

We also designed another construct aimed at incorporating a significant portion of the MT domain, to enhance the replication rate of CGMMV genome-based deletion constructs. This design divided the entire CGMMV genome sequence into three major segments: the N′-terminal part containing the 5′ UTR, with 87% of the MT domain (1–1173 nt); a central part with a putative MP-subgenomic promoter ranging from 4900 nt to 5080 nt; and the C′ terminal part carrying CP coding sequences with the CP-SGP and 3′ UTR (5612 nt to 6424 nt) of the genome. These segments were joined together to form DG-2, resembling a miniature CGMMV (Figure 2). Based on the secondary structure analysis of the complete CGMMV genome using various Web tools, we identified consensus regulatory sequences within the 1–1123 nt region of the N-terminal end (Appendix A). To preserve these sequences, we retained 1173 nt, equivalent to 371 amino acids of the MT domain. Additionally, we selected 90 nt upstream and downstream of MP-TSS (+1), based on in silico predictions of MP-SGP sequences. The core MP promoter was retained for functional genomic studies. Finally, a long stretch of genomic segments from the 3′ terminal end (5612–6424 nt), containing the CP coding sequences along with 150 nt upstream, was retained to maintain the full functionality of the CP promoter. We also included conserved nucleotides, necessary for the formation of pseudoknots and t-RNA-like structures in the 3′ UTR, without any modification, as the structural conservation of the 3′ UTR is necessary for the binding of the RdRp domain and subsequent genome replication. Furthermore, we incorporated several restriction sites within the DG-2 genome sequence to enhance its flexibility for various applications. A 714 nt sequence of the enhanced green fluorescent protein (eGFP) gene was included after 105 nt from CP-TSS (+1), which was previously identified as the active core promoter length of the CP. Ultimately, an ORF search of this deconstructed genome revealed the presence of three functional overlapping ORFs: ORF1 (1245 nt), encoding 98% of the Methyltransferase domain; ORF2 (276 nt), encoding 68% of the MP; and ORF3 (1239 nt), encoding the CP fused with the eGFP (412 amino acids). A detailed genome map (2961 nt) of DG-2 is presented in Figure 2. This sequence was synthesised, together with a 2 × 35 s promoter, a ribozyme (Rz) site, and a nopaline synthase (Nos) terminator, through outsourcing from Bio Basic Canada Inc.

The synthetic genome construct (4062 nt) containing the 2 × 35 s promoter, DG-2 genome sequence, ribozyme (Rz) site, and Nos terminator was excised from the pUC57 vector (2.7 Kb) backbone using KpnI and SacI restriction sites (Appendix A), then directly cloned into pGreen II (0029) using the same restriction sites. This removal eliminated the entire multiple cloning sites (MCS) from the pGreen II (0029) backbone (4632 bp), allowing us to utilise these sites for the DG-2 genome modification. The positive clones from the colony PCR (Appendix A) were further confirmed through restriction digestion using the BamHI and HindIII restriction enzymes, releasing the desired 2.78 kb fragments from the vector (4.7 Kb) backbone (Appendix A). The confirmed plasmid constructs were then transferred into Agrobacterium GV-3101, along with the pSoup helper plasmid, via electroporation. Co-transformation of pGreen and pSoup was performed using equal concentrations of both plasmids, with only kanamycin (no tetracycline) and rifampicin used for selective screening of the transformed Agrobacterium colonies. Positive colonies were confirmed through a colony PCR (Appendix A) and subsequently maintained for further use.

### 3.2. Trans-Replication and In Planta Systemic Movement of Deconstructed Genomes of CGMMV 

#### 3.2.1. Trans-Replication and In Planta Systemic Movement of DG1

Initially, we used terminal primers (486F and 489R) to detect the replicating dRNA of DG-1 and studied its replication behaviour in infiltrated *N. benthamiana* leaves at different time intervals. The replication was not detected in plants when DG-1 was infiltrated alone, indicating the loss of infectivity and self-replicating ability of DG-1 (Appendix A). The replicon was trans-replicated only when DG-1 was infiltrated into CGMMV-infected leaves. Interestingly, the DG-1 replicon was not identified when co-infiltrated with wild-type CGMMV, suggesting that CGMMV initially does not support the replication of DG-1 after co-inoculation (Appendix A). To validate this, we repeated the experiment. Furthermore, we infiltrated the DG-1 plasmid construct into *N. benthamiana* 7 days after the initial CGMMV infiltration. In this case, the replicon was identified both in the infiltrated leaf and the systemic leaf, but its accumulation within the infected tissue varied significantly, possibly due to the disruption of the movement protein. Both symptomatic and asymptomatic strains of CGMMV acted as helper viruses for replicating DG-1 in *N. benthamiana* (Figure 3a). In contrast, the trans-replication of DG-1 was not evident in CGMMV-infected cucurbitaceous hosts such as bottle gourd, cucumber, and watermelon (Appendix A), where it was not supported by the wild types of CGMMV (Appendix A). The lack of proper delivery methods may be the reason, and further clarification is necessary.

Furthermore, the authors tried to develop a specific duplex PCR-based detection system for the simultaneous detection of DG-1, along with its helper CGMMV, and to avoid the amplification of multiple fragments that had been encountered during the use of terminal primers (i.e., BM-486F and BM-489R) (Figure 3a). For that, a new primer set was designed through the combination of three primers, viz., BM-1171F, BM-1182F, and BM-1173R. BM-1171F and BM-1173R were used to detect the helper CGMMV amplifying the 1415 bp long amplicon, whereas BM-1172F with BM-1173R was used for the specific detection of the DG-1 replicon (Figure 3b). BM-1171F did not bind to DG-1, as its coordinates (4591–4616 nt) were deleted from the CGMMV; whereas, BM-1182F is able to bind to wild type also, but it generated 5204 nt amplicons (for BP4) that can be avoided by restricting the extensions time of the PCR within 1 min. 

To confirm the replication of DG-1 within the infiltrated tissue, we examined its replication rate at different time intervals and observed an increasing trend in replication, using a semi-quantitative PCR. The DG-1 replicon reached a detectable level within 48 h of infiltration, with its concentration increasing and plateauing within the 5th to 6th days post-infiltration (DPI), before either stabilising or gradually declining. Subgenomic RNAs and other progeny RNAs from the wild-type genome remained nearly constant during this period, as shown in Figure 4a. This indicated the limited influence of DG-1 on the functionality of wild-type CGMMV. Furthermore, the in planta systemic movement of the DG-1 replicon was determined. We tested different leaves of *N. benthamiana*, positioned upward from the infiltrated leaf (L0). These leaves were designated as L1 to L10, based on their relative positions. RNA was extracted separately from these individual leaves at 14 DPI, and trans-replication and movement assays of DG-1 were conducted using an RT-PCR with primer pairs BM-486F and BM-489R, which generated an approximately 2.0 kb amplification product, along with an amplification of sub-viral genomic fragments. We detected the DG-1 replicon (approximately 2.0 kb) in all the studied leaves (L1 to L10) of *N. benthamiana* (Figure 4b). This systemic movement of the DG-1 replicon was supported by both symptomatic (BP4) and asymptomatic (BP7) CGMMV helper viruses. DG-1 was observed to move from the infiltrated leaf (0 L) to the first systemic leaf (1 L) within 2–3 DPI and subsequently reached the 2nd systemic leaf after 3 DPI. It continued to spread systematically to different parts of the plant. However, the local cell-to-cell movement within a systemic leaf appeared uneven, as evidenced by the day-wise distribution pattern in the first and second systemic leaves. Thus, it is plausible that the movement protein of helper CGMMV may not uniformly facilitate the distribution of its deconstructed replicon within the infected tissue of the leaf. The experiments were repeated thrice, and a total of six plants per replication were tested by RT-PCR. DG-1 was detected in all 18 plants that were infected with either symptomatic (pBP4) or asymptomatic (pBP7) CGMMV infectious genome construct (Appendix A). Thus, it was concluded that both the symptomatic (BP4) and asymptomatic (BP7) CGMMV were efficient in supporting DG-1 in its replication and movement; whereas, DG-1 alone was unable to self-replicate.

#### 3.2.2. Trans-replication and In-Planta Systemic Movement of DG2

To predict the trans-replication of the DG2 construct with the help of CGMMV, we used an RT-PCR to detect infiltrated leaf samples using GFP-specific primers (BM-1265F and BM-1266R). DG-2 was detected within the infiltrated tissue but was not detected in leaves infiltrated with DG2 alone or CGMMV alone (Figure 5a). We infiltrated DG2 at 7 days post-infection (DPI) from wild-type CGMMV constructs (BP4/BP7), eliminating the possibility of the self-replication of DG-2 without the support of wild-type CGMMV. Further, to investigate long-distance systemic movement, a similar RT-PCR was performed, using both infiltrated and systemic leaf samples; it showed the presence of DG-2 within infiltrated tissue but its absence in the first systemic leaves (Figure 5b), thus indicating the restricted systemic movement of DG-2 in *N. benthamiana*. Further, the experiment was repeated with symptomatic, as well as asymptomatic, CGMMV; a similar result was obtained. Finally, to prove its trans-replication within infiltrated tissue, a gradual increase in the replication rate was visualised over time when tested at a 2-day interval (Figure 5c), keeping pCambia 1302 as a positive control. The trans-replication of DG-2, with its restricted systemic movement, provided a new dimension to our research.

### 3.3. Translational Ability of Deconstructed Genome-Based Bipartite Vector System in CGMMV-Infected N. benthamiana

#### 3.3.1. Functionality of Deconstructed Virus Genome-Based Silencing Vector

##### Development of Deconstructed Virus Genome-Based Silencing Vector

To assess the gene-silencing efficiency of the deconstructed virus genome (DG), we inserted a partial fragment of the *NbPDS* gene into the DG-1 plasmid construct in the sense orientation, positioned after the CP coding frame. We amplified a small partial sequence (227 bp) of the *NbPDS* gene from its mRNA transcript and inserted it into the genome using the LC- PCR strategy [20]. This process involved two rounds of PCR. In the first round, we used primers to amplify the desired *NbPDS* gene fragment, while retaining overhang flanking regions (P1 and P2). P1 was homologous to the target region of the vector sequence, while P2 was complementary to the subsequent region of the viral vector. The purified PCR amplicons were then used as primers for the second PCR, in which the overhang parts (P1 and P2) facilitated the specific binding of the insert to the virus vector. In this PCR, a third primer (P3) was added, which acted as a reverse primer containing a few complementary sequences of the joining ends of the insert and vector. This facilitated the amplification of the desired VIGS vector, which contained the PDS gene of interest incorporated in the final PCR product as a circular PCR molecule (PDS + vector) with nicked ends. This circular PCR molecule contained a circular plasmid that was treated with the DpnI enzyme, resulting in the degradation of plasmid molecules and the transformation of the circular PCR molecule into *E. coli*-competent cells, leading to the appearance of a few colonies. All colonies were further confirmed by a colony PCR, resulting in all positive colonies (Appendix A), and were further validated through restriction digestion of the plasmid with BamH1 and XbaI restriction enzymes (Appendix A). Upon restriction digestion, a 2.227 kb fragment was released from pDG(NbPDS)-1, which was comparable in size to the 2.0 kb size of pDG-1. The newly generated construct was named pDG(NbPDS)-1(Appendix A).

##### Phenotypic Silencing of NbPDS Gene Using pDG(PDS)-1 Construct in CGMMV-Infected *N. benthamiana*

The pDG(PDS)-1 construct was agroinfiltrated into CGMMV-infected *N. benthamiana* plants in the early growth stage, typically when they had two leaves. This resulted in the development of a photobleaching phenotype in approximately 30% of the plants, characterised by phenotypic changes that began around 15 DPI and continued up to 45 DPI (Appendix A). Initially, small yellowish-white spots appeared on the leaves and gradually expanded, with most of the photobleaching occurring around the veins. Although these phenotypic changes were visible on only a few leaves of each plant, they appeared randomly throughout the top leaves, indicating the long-distance systemic movement of the DG(PDS)-1 transcript. The frequency and intensity of gene silencing were higher when symptomatic CGMMV (pBP4) was used as the helper, compared to asymptomatic CGMMV (pBP7) as the helper virus (Appendix A). This difference suggested a potential variation in the replication rate of pDG(PDS)-1 with its helper CGMMV. These results provided phenotypic evidence of the functional significance of the pDG-1 construct as a VIGS vector.

##### Detection of Replication and Movement of DG(PDS)-1 in CGMMV-Infected *N. benthamiana*

To confirm the replication and movement of DG(PDS)-1 with the help of wild-type CGMMV in *N. benthamiana*, we detected the production of RNA replicons from the DG(PDS)-1 construct using an RT-PCR. We also assessed the replication rate of DG(PDS)-1 in comparison to its helper CGMMV within infiltrated tissue, using a duplex PCR approach. One primer set (96F and 1268R) was used for the specific amplification of helper CGMMV, while another primer set (1180F and 1269R) was specifically for DG(PDS)-1. DG(PDS)-1 was specifically detected by amplifying the CP fused with the PDS fragment, using the CP forward primer (BM-1180F) and PDS reverse primer (BM-1269R). DG(PDS)-1 was detected within the 0 to 9 DPI period (Figure 6a), but the intensity of the amplicon was significantly lower than that of the well-established helper CGMMV. Furthermore, over time, the replication rate remained stable up to 7 DPI, after which, it gradually decreased, indicating the pattern of trans-replication of DG(PDS)-1 with respect to its helper CGMMV. As per our observation, DG-1 replicated within the infiltrated tissue. Its replication was initiated, and it systematically moved from the infiltrated leaf (0 L) to the first systemic leaf (1 L) within 2–3 DPI (Figure 6b), subsequently reaching the second systemic leaf after 3 DPI (Figure 6c). In this manner, it was systematically distributed within different parts of the plant, as evidenced by the photobleaching of leaves. However, the local cell-to-cell movement within a leaf lamina appeared uneven, as visualised by the day-wise distribution pattern in the first and second systemic leaves. This suggests that the movement protein of helper CGMMV may not facilitate the uniform cell-to-cell local distribution of the deconstructed replicon within the infected tissue of the leaf.

##### Subgenomic RNAs of Deconstructed Genome DG(PDS)-1

To examine whether the replication of DG-1 leads to the production of subgenomic RNAs (sgRNAs) similar to wild-type CGMMV, we conducted a Northern blot analysis using specific probes targeting the sgRNA regions of CGMMV. The GC-rich 3′ terminal of CGMMV was used in the preparation of 225 nt long probes. To determine the probe sensitivity, the total RNA was diluted in different concentrations and a dot blot was performed, and the probe sensitivity was detected up to a picogram concentration. The same probe was used for detecting the DG(PDS)-1 replicon, along with viral gRNA and sgRNA. The replicon was detected in the infiltrated, as well as in the first systemic, leaf of *N. benthamiana*, proving the replicating nature of DG(PDS)-1 with the help of wild-type CGMMV. The expression of DG(PDS)-1 is very limited in comparison with its helper genome. The excessive expression of two sgRNAs was evident in the Northern blot (Figure 7).

#### 3.3.2. Functionality of Deconstructed Virus Genome-Based Protein Expression Vector

##### Protein Expression Ability of Deconstructed Genome 2 (DG2)

The CGMMV-infected *N. benthamiana* plants that were agroinfiltrated with the DG2 construct exhibited colour changes in the infiltrated tissue. When we examined these infiltrated tissues under confocal microscopy, we observed the expression of GFP. The harvested tissue from the infiltrated area at different time points showed the expression of GFP, starting from 2 DPI and extending up to 12 DPI or more (Figure 8A). A prominent expression of GFP was visualised at 6–8 DPI, after which it gradually declined but remained detectable up to 12 DPI or more (Figure 8B). Interestingly, the expression of GFP within the leaf was scattered and restricted to specific tissue, indicating that the DG-2 replicon was limited to the specialised non-vascular tissue of the *N. benthamiana* leaf and unable to cross certain barriers, restricting its long-distance movement.

##### Confirmation of Protein Expression Ability of Deconstructed Genome 2

To confirm the protein expression ability of DG2, we performed a Western blot analysis, using protein extracts from *N. benthamiana* leaves infiltrated with DG-2 alone, with CGMMV (BP4) as a helper virus. The analysis was conducted at different time points (2, 4, 6, and 8 DPI). A GFP-specific antibody was used to detect the expression of GFP, which is encoded by the pDG-2 construct. The Western blot assay of the extracted total crude protein exhibited a similar expression pattern of GFP. The in planta expression of GFP was detected using GFP polyclonal antibody, confirming GFP expression starting from 2 DPI and extending up to 15 DPI (Appendix A). The highest expression of GFP was visualised during the 6–8 DPI time period (Figure 9), after which, the GFP expression level slowly declined but remained detectable using the GFP-specific antibody. Here, the GFP expression using pCambia 1302 was used as the control. Notably, the GFP expression level of DG2 was significantly higher than that of pCambia 2300, indicating its translational ability to express foreign protein in planta. The expression of GFP was observed at all time points tested, indicating the successful expression of the protein from the DG-2 replicon.

### 3.4. Quantification of Replication Rate of the Deconstructed Genomes

To assess the replication pattern of the DG-1 replicon, we conducted a relative quantification using a qRT-PCR, with host actin (*NbActin*) as the reference gene. Simultaneously, we quantified the relative abundance of helper CGMMV (BP4), to investigate the impact of DG-1 on its helper virus. We extracted RNA from 5 mm leaf discs obtained from a fixed infiltrated leaf of *N. benthamiana* at different time points (0 to 12 DPI) and subjected it to a reverse transcription to prepare cDNA. The qRT-PCR was conducted using primers specific to DG-1 and BP4, along with the reference *NbActin* gene. The ΔΔCt method was employed to determine the fold change in the abundance of DG-1 and BP4 compared to the reference gene (*NbActin*). The qRT-PCR results revealed that the DG-1 replicon initially had a low level of replication in the infiltrated tissue at 0 DPI. However, its replication rate increased significantly and reached its maximum level by 5 DPI (Appendix A). Subsequently, the replication rate of DG-1 decreased but remained relatively constant up to 9 DPI. These results corroborated the findings from the semi-quantitative PCR and indicated that the DG-1 replicon effectively replicates and persists within the infiltrated tissue. Additionally, we found that DG-1 did not significantly impact the replication of helper CGMMV, as the abundance of BP4 remained relatively constant throughout the time points studied. 

For DG2, we performed a qRT-PCR, using GFP-specific primers (BM-1265F and BM-1266R) and actin as a reference gene, to quantify the expression of GFP within the infiltrated tissue at different time intervals. The results revealed an increasing trend in the replication rate of DG-2 within the infiltrated tissue, with the highest level of GFP expression observed at 8 DPI (Appendix A). This confirmed the replication and expression of the DG-2 construct and its ability to express GFP in planta.

### 3.5. Genome Sequence Analysis of DG-1 and DG-2

The Sanger sequencing method was employed to determine the complete genome sequences of DG-1 and DG-2. An analysis of these sequences was conducted to validate the integrity and precision of the deconstructed genomes. Furthermore, sequence comparisons were executed to detect any potential mutations or variations in comparison to the wild-type CGMMV genome.

#### 3.5.1. Genome Sequence Analysis of DG-1

The determination of the complete genome sequence of DG-1 is illustrated in Appendix A. A sequence analysis affirmed the faithful representation of the deconstructed genome, with no observed mutations or variations relative to the wild-type CGMMV genome. This underscores the high fidelity of the DG-1 construct.

#### 3.5.2. Genome Sequence Analysis of DG-2

The determination of the complete genome sequence of DG-2 is depicted in Appendix A. A sequence analysis validated the precise representation of the deconstructed genome. No mutations or variations were detected in comparison to the wild-type CGMMV genome, affirming the high fidelity of the DG-2 construct.

## 4. Discussion

The emergence of defective genomes during virus pathogenesis is a natural phenomenon. These defective genomes are spontaneously generated during the replication of a virus, especially those with RNA genomes [27], particularly when their concentration reaches a high titer [26]. It is hypothesised that the premature detachment of RdRp from its negative-strand template during replication (the breakpoint), and the subsequent reattachment to the same template, lead to the synthesis of defective genomes [28,29,30]. Although these genomes are defective in terms of self-replication and pathogenesis due to the absence of substantial portions of their parental viral genome, they still carry some of the key regulatory genomic elements essential for replication. Consequently, these truncated forms can only replicate with the assistance of wild viruses, which act as helpers by providing replicase enzymes. They may either interfere with the replication of the parental virus [31] or not, hence they are referred to as defective viral genomes (DVGs) [32,33]. They have a significant functional impact on the pathogenesis and evolution of the wild-type virus. Naturally, their occurrence has been reported during late infection events of many human and animal viruses, such as dengue virus [34], Ebola virus [35], hepatitis C virus [36], influenza A virus [37,38], mumps virus [39], poliovirus [40], respiratory syncytial virus [41], Sendai virus [42], Sindbis virus [43], etc. Similar subviral agents are also evident [44] in the case of many plant viruses like Bromovirus [45], Carmovirus [46], Flexivirus [47], Tombusvirus [48], etc., which are synthesised de novo [49]. However, such a documentation of naturally occurring DVGs is lacking in the case of the majority of Tobamoviruses [50]. This raises questions about the efficacy of CGMMV in supporting the trans-replication of subgenomic virus particles. Therefore, different artificial forms of dRNAs were generated from their genome to study their functional significance.

Previously, DVGs were artificially generated from the full-length RNA genome of TMV by deleting internal genome sequences [24,50,51] that contain cis-regulatory elements for replication and movement. Building on this information from TMV, we attempted to develop a deconstructed genome called DG-1 derived from the CGMMV genome, and its functionality was tested. The DG-1 construct of CGMMV lost its ability for self-replication and in planta movement due to the large internal deletion of the genome, including the helicase, RNA-dependent RNA polymerase domain of the replicase enzyme, and a significant part of the movement protein. As a result, the DG-1 replicon was not detected when infiltrated alone, neither in the infiltrated leaves nor in the systemic leaves of *N. benthamiana*, consistent with the previous findings of Ogawa et al. [52], where large deletions of the genes encoding the methyltransferase, helicase, and RdRp domains of replicase generated replication-defective mutants in TMV. DG-1 can only replicate and move systemically within *N. benthamiana* with the help of wild-type CGMMV. In this scenario, wild-type CGMMV provides the replicase enzyme to support the replication of DG-1, similar to the earlier experiments by Raffo and Dawson [53] in TMV. They reported that artificially created subgenomic replicons—named KL, which retained 1–256 nt of the 5′ terminal and 5237–6395 nt of the 3′ terminal ends, and KLL, containing only 1–258 nt of the 5′ terminal ends and 4900–6395 nt of the 3’ terminal ends of the TMV genome—could replicate and spread systemically within tobacco plants once co-inoculated with wild-type TMV [53,54]. This supports our hypothesis that wild-type CGMMV can assist in the trans-replication of subviral particles. However, the DG-1 replicon (dRNA) was not detected in the infiltrated leaves of *N. benthamiana* when co-infiltrated with wild-type CGMMV (either symptomatic BP4 or asymptomatic BP7). This differs from previous findings for TMV, where the trans-replication of dRNAs was reported when co-inoculated with parental TMV, either in tobacco protoplasts or in the leaves of *N. benthamiana* plants. This difference might be related to the methods of delivery in the plant system. Previously, in vitro transcripts of dRNA genomes were used for the inoculation of protoplasts, and after 1–2 days of growth in protoplasts, the cell lysate containing the active viral culture was directly inoculated into the plant leaves, allowing for the rapid multiplication of dRNAs along with their helper virus. In contrast, during co-agroinfiltration of the cDNA construct of both DG-1 and ‘helper’ CGMMV, the wild-type virus does not sufficiently support the amplification of DG-1 dRNA transcripts, due to the lack of sufficient replicase enzymes in planta. In the initial stage of infection by CGMMV, it produces replicase to replicate its own genome for the establishment of infection and colonisation. It requires time to replicate its own genome up to a critical level at which it will support the replication of other dependent genomes, such as the replication of dRNA immediately after infection. Identifying this critical time is necessary for achieving a sufficient amplification of DG-1, which may require further refinement in the delivery system.

Naturally occurring dRNAs possess certain cis-acting elements that are necessary for efficient replication and movement with the help of the parental virus. This principle was applied to generate DG-1 based on previously reported information on TMV dRNAs. DG-1 contains 1–834 nt of the N-terminal part and 5271 to 6424 nt of the C-terminal part of the genome, based on the genome map of the deconstructed RNA (∆HINC151) of TMV [24], which retained 1–841 nt (258 aa) of the N-terminal and 5182–6395 nt of the C-terminal of its genome. The evidence of in planta trans-replication and systemic movement of DG-1 with the help of wild-type CGMMV is in accordance with the ∆HINC151 of TMV. Interestingly, the ∆HINC151 of TMV is also reported to accumulate in the inoculated leaves of *N. benthamiana* about 4–5 times more highly than its helper TMV [24]. However, in our case, DG-1 is found to replicate at a maximum of 4.5 times over actin, which is about 7.7 times lower than wild-type CGMMV. After the cellular accumulation of DG-1 within infected tissue, DG-1 can spread systemically up to the tenth leaf or even more (as per our observation) in *N. benthamiana* via vascular bundles, which is not evident in the case of the ∆HINC151 of TMV, which is reported to move within specialised non-vascular tissue, thus restricting its systemic movement [24]. By contrast, the DG-1 replicon can systemically move long distances throughout the plant, but its uniform distribution in all local tissue is limited. This suggests that the 258 aa of the MT domain of CGMMV is necessary for the long-distance movement of dRNA but has a negative impact on its higher accumulation. This finding is consistent with the previous report of Knapp et al. [24,51] on TMV, where they elucidated the functional significance of the N-terminal MT domain containing (1–258 aa) in virus movement [24]. Furthermore, this MT domain also plays a crucial role in the formation of a methylated cap that enhances the stability of viral genomic RNA. Interestingly, this 5′ RNA capping mechanism is conserved among diverse members of the Alphavirus-like superfamily, despite their limited amino acid identity [53]. A comparative analysis of the MT domain of various plant-infecting viruses reveals some conserved amino acid residues, including histidine (H81), aspartate (D134), arginine (R137), and tyrosine (Y271), located at its N-terminal part, forming a catalytic tetrad that might have a significant role in cap formation [55]. Disruption of any one of these residues greatly hampers the methyltransferase and AdoMet-dependent guanylyltransferase activity of this enzyme, affecting RNA capping [56] and resulting in the subsequent instability of newly synthesised RNA, as observed in the Semliki Forest virus [57] and bamboo mosaic virus [58]. The existence of a partial N-terminal domain in the DG-1 of CGMMV may also impair RNA stability. Thus, despite its replication, the limited stability of the DG-1 replicon affects its higher accumulation.

To address the shortcomings of DG-1, DG-2 was created by incorporating 1173 nt (371 aa) of the MT domain, with the aim of achieving higher replication. Additionally, multiple cloning sites were incorporated under the control of subgenomic promoters, allowing for the simultaneous expression of multiple foreign genes. When the DG-2 construct was infiltrated into CGMMV-infected *N. benthamiana*, DG-2 was trans-replicated with the help of wild-type CGMMV. Detection was carried out using an RT-PCR, but DG-2 was not detected within the systemic leaf tissue. This restricted movement of the deconstructed genome was previously reported for TMV [51], where two categories of dRNAs were identified: ones that could not move but were replicated by the wild-type virus and others that could move but were not replicated by the wild-type virus. RT-PCR-based detection showed that the replication rate of DG-2 gradually increased but remained restricted within the infiltrated tissue. This indicates that the C’-terminal part of the MT domain (i.e., 258–371 aa) is involved in the higher accumulation of dRNAs, possibly due to the proper capping mechanism. Furthermore, this region may also be involved in the restricted movement of dRNAs. Similar evidence was reported by Knapp et al. [24], demonstrating the higher accumulation but restricted movement of the deconstructed RNA genome, ∆Cla151, which possessed the 425 aa of the MT domain. Previously, the role of a non-conserved region located between the MT domain and helicase domain in cell-to-cell movement was reported, and the mutation of certain nucleotides in the 248 aa to 538 aa region hampered this function in TMV [59]. From the confocal microscopic analysis of DG-2 in the infiltrated tissue, it was evident that the localisation of DG-2 was scattered within the lamina. Therefore, it could be predicted that the cell-to-cell movement of DG-2 might be restricted due to the extensive modification of the MT domain. Further, a qPCR assay of the DG-2 replicon indicated an increase in its replication rate over time (within 15 DPI), from 0.067 to 0.098 ng/g tissue. The quantification of GFP expression within the infiltrated tissue also indicates that trans-replication of DG-2 is assisted by wild-type CGMMV. Apart from its trans-replication, DG-2 also exhibits better translational ability. To demonstrate this, the GFP protein was inserted within the coding frame of the CP and was successfully expressed within infiltrated tissue. The expression of GFP was visualised over time in confocal microscopy, showing that GFP expression began on the 2nd DPI and reached a high level between the 6th and 10th DPI, gradually decreasing but remaining expressed up to the 15th DPI. This finding was also supported by Western blot assays, where the amount of GFP translation in the form of the CP-fused protein (48 kDa) was quantified using polyclonal anti-GFP antibodies. The pattern of GFP expression was consistent with the confocal visualisation, confirming that DG-2 can be used to express a foreign protein in plant tissues. Therefore, DG-2 can be utilised to establish a bipartite vector system using the CGMMV genome. A similar bipartite system based on TMV and its dRNA was developed by Knapp et al. [51], which was later explored as a novel two-component vector system (dRT-V) for the high-level expression of multiple therapeutic proteins, including a human monoclonal antibody, in plants [14]. This new design based on the deconstructed genome of TMV proved to be a successful alternative. In this system, a defective/deconstructed RNA (dRNA) of TMV was created by a major deletion of replicase and movement protein, and its functionality (trans-replication and movement) was demonstrated using a helper TMV construct. Importantly, two foreign genes were inserted, one in the helper construct and one in the deleted construct, enabling simultaneous expression in the same cell. This non-competitive bipartite module is named the defective RNA (dRNA)-based TMV vector (dRT-V), and it is used for the expression of mAbs IgG against the antigen of Bacillus anthracis in *N. benthamiana* at 120 mg/kg FWT. Similarly, our bipartite system can also be exploited for the expression of multiple proteins in plants.

In conclusion, two novel two-component CGMMV-based vector systems have been developed (Figure 10). The DG-1 replicon can replicate *in trans* with the assistance of its helper CGMMV but exhibits lower stability, reducing its accumulation in high copy numbers. However, it can move cell to cell and cross the vascular bundle barrier to reach the upper leaves of *N. benthamiana*. In this way, DG-1, along with helper CGMMV, represents a bipartite model system with functional similarities to the multipartite genome systems of other plant viruses. In planta functional validation of these bipartite systems in CGMMV demonstrates their utility as gene-silencing vectors for deciphering plant functional genomics. In the second system, DG-2 replicates at a much higher rate but remains localised within the infiltrated tissue. It also exhibits a translational ability for foreign proteins, as demonstrated through the expression of GFP within *N. benthamiana*. Thus, the creation of a bipartite system using the CGMMV genome has been successfully achieved.

## Figures and Tables

**Figure 1 plants-13-01414-f001:**
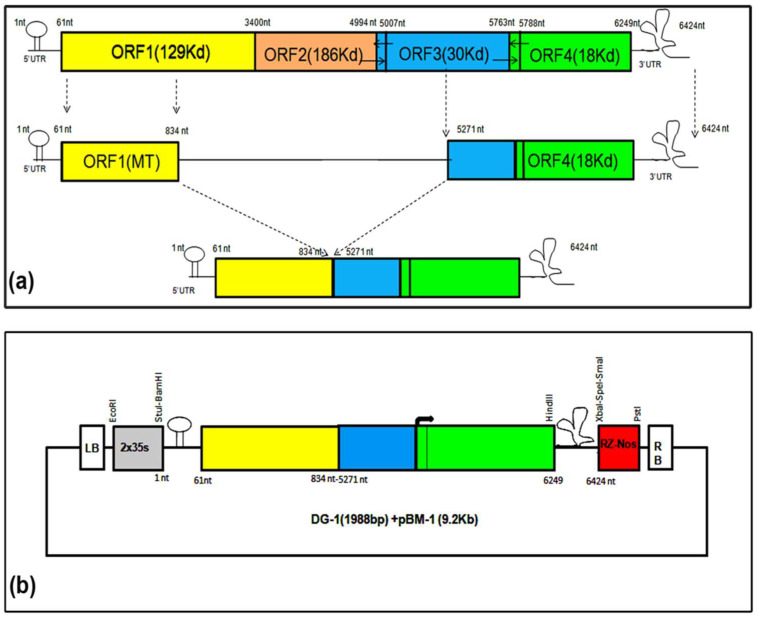
Designing of deconstructed genome-1 of CGMMV. It was designed based on the genome organisation of dRT-V from TMV (Roy et al., 2010). It contains 1–834 nt of the 5′ terminal and 5271–6424 nt of the 3′ terminal of the genome. (**a**) DG-1 (1988 nt) was created through the PCR-based deletion of the BP4 infectious clone of CGMMV. A significant part of the replicase and movement protein was removed. (**b**) DG-1 was cloned into a modified binary vector, pBM1, containing the 2 × 35 s promoter after the left border (LB) and ribozyme sites (Rz), followed by a Nos terminator just before the right border (RB). Two-step PCR with high-fidelity Taq polymerase was used for creating DG-1.

**Figure 2 plants-13-01414-f002:**
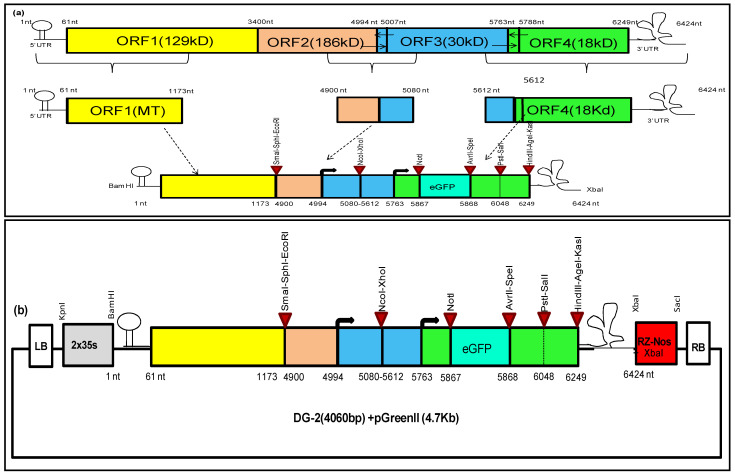
Designing and cloning of deconstructed virus genome-2 of CGMMV. (**a**) Design of deconstructed genome-2 based on CGMMV full genome. The deconstructed genome-2 was designed based on our in silico prediction. It contains 1173 nt of the 5′ terminal and 5612 nt of the 3′ terminal of the genome, with an internal fusion of the putative MP–subgenomic promoter region (4900–5080 nt). (**b**). Cloning of deconstructed genome-2 of CGMMV into pGreenII vector was performed, along with 2 × 35 s promoter, in upstream of 5′ terminal end and ribozyme (Rz) site–Nos terminator in downstream of 3′ UTR. The entire cassette was inserted in between the left border (LB) and right border (RB) of the pGreen II backbone with KpnI and SacI restriction sites.

**Figure 3 plants-13-01414-f003:**
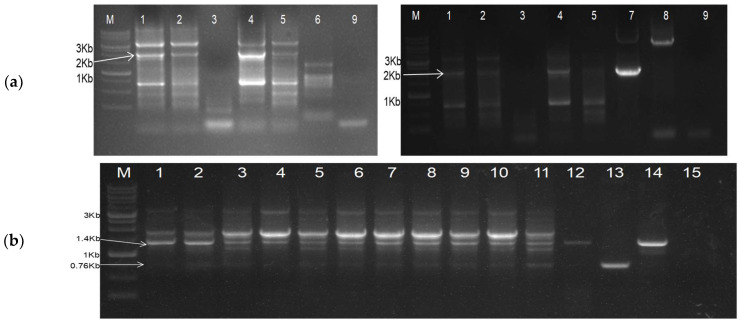
RT-PCR based detection of DG-1 constructs, when infiltrated into CGMMV-infected *N. benthamiana* (**a**) Amplification of DG-1 replicon from CGMMV-infected *N. benthamiana* using primers BM-486F and BM-489R. Lanes 1 and 2 represent the DG-1-infiltrated leaf and the first systemic leaf of BP4 infected *N. benthamiana*. Lane 3 is the only DG-1-infiltrated leaf of healthy *N. benthamiana*. Lanes 4 and 5 represent the DG-1-infiltrated leaf and the first systemic leaf of BP7-infected *N. benthamiana*. Lane 6 is the only BP4 (CGMMV)-infected *N. benthamiana*. Lane 7 is the DG-1 plasmid control, Lane 8 is the BP4 plasmid control, and Lane 9 is the reagent control (negative). (**b**) Duplex PCR-based detection of DG-1 replicon, along with helper CGMMV, when DG-1 is infiltrated into CGMMV-infected *N. benthamiana.* The RT-PCR protocol was standardised by using a combination of three primers: BM-1171F, BM-1172F, and BM-1173R. Lanes 1 to 11 represent the co-existence of the DG-1 replicon along with BP4 in different *N. benthamiana* plants. Lane 12 is the only BP4-infected *N. benthamiana*. Lane 13 is the pDG-1 plasmid control; lane 14 is the pBP4 plasmid control; and lane 15 is the negative (reagent) control.

**Figure 4 plants-13-01414-f004:**
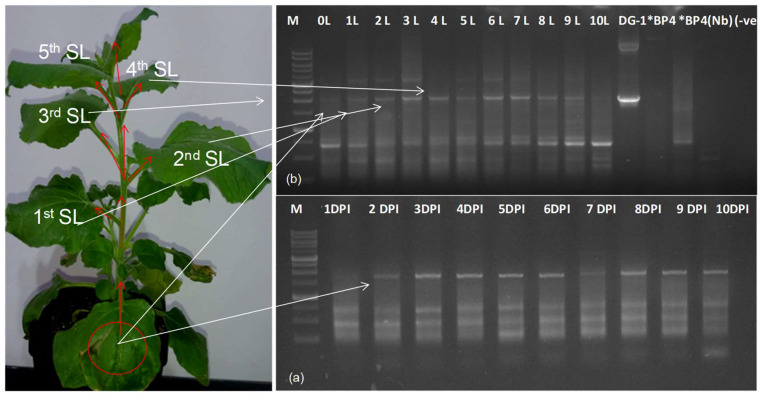
Trans-replication and systemic movement of DG-1 replicon in CGMMV-infected *N. benthamiana*. (**a**) The replication of DG-1 in the infiltrated leaf (0 L) of CGMMV-infected *N. benthamiana* showed a gradual increment in the concentration of DG-1 up to 6DPI; after which, its concentration either remain stable or decreased gradually. (**b**) After replication, the DG-1 replicon moved systemically into the systemic leaves, and then subsequently spread into the whole plant. It was tested up to the 10th SL of *N. benthamiana*. Here, *BP4 represents the plasmid construct and *BP4(Nb) represents BP4 infected *N. benthamiana* plant as control.

**Figure 5 plants-13-01414-f005:**
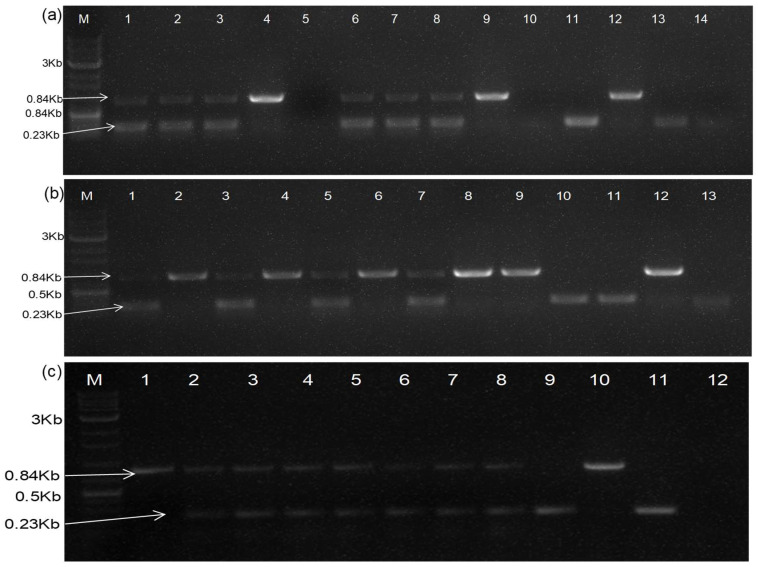
Duplex RT-PCR-based detection of deconstructed genome-2 within CGMMV-infected *N. benthamiana,* using two primer sets; BM-96F and BM-1268R are used to predict infection from helper CGMMV, whereas BM-1265F and BM-1266R are used to detect DG-2 replicon. (**a**) The presence of the DG-2 replicon within infiltrated tissue signifies its trans-replication with the help of CGMMV, as it was not detected from the only-DG2-infiltrated tissue (in the absence of CGMMV) as well as only-CGMMV-infected tissue (in the absence of DG2). Here, lanes 1–3 are the symptomatic (BP4) CGMMV-infected *N. benthamiana* plants agroinfiltrated with DG2, lanes 4 and 5 represent only-BP4- and DG2-infiltrated leaf tissue, respectively. Lanes 6–8 are the asymptomatic (BP7) CGMMV-infected *N. benthamiana* plants agroinfiltrated with DG2, lanes 9 and 10 represent only-BP7- and DG2-infiltrated leaf tissue, respectively. Lanes 11, 12, and 13 depict the DG2 plasmid control, BP4 plasmid control, and pCambia 1302 plasmid control, with a negative control in lane 14. (**b**) To check the long-distance systemic movement of DG2, a similar RT-PCR was performed using RNA isolated from infiltrated and systemic leaf samples; the gel picture showed the presence of DG-2 within infiltrated tissue (lanes 1, 3, 5, 7) but its absence in the systemic leaf tissue (lanes 2, 4, 6, 8), as well as in the BP4-infected plant (lane 9). Here, lanes 10, 11, and 12 are the pCambia 1302 plasmid control, DG2 plasmid control, and BP4 plasmid control, respectively. (**c**) The gradual increment in the replication rate of DG2 was visualised over time when tested at 0 DPI (1), 2 DPI (2), 4 DPI (3), 6 DPI (4), 8 DPI (5), 10 DPI (6), 12 DPI (7), and 15 DPI (8), and compared with pCambia 1302 at the 5th DPI (9). Here lanes 10 and 11 are the BP4, and DG2 plasmid control, with the reagent control in lane 12.

**Figure 6 plants-13-01414-f006:**
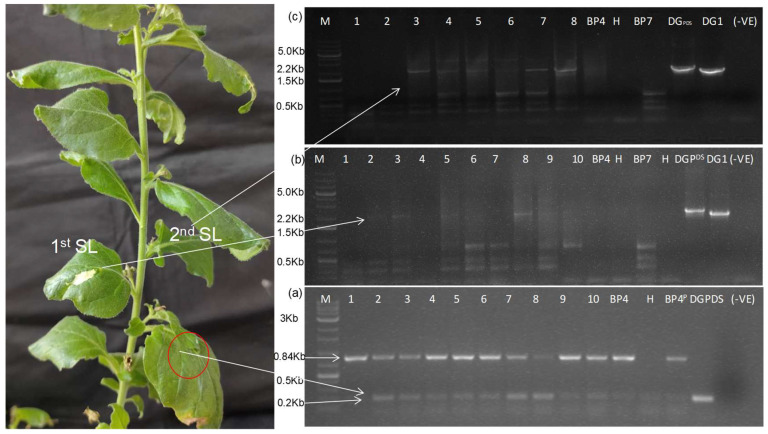
The trans-replication and movement of DG(PDS)-1 with the help of wild-type CGMMV in *N. benthamiana*. (**a**) Duplex PCR-based detection of CGMMV and DG (PDS)-1 replicon in infiltrated tissue of *N. benthamiana*. The RT-PCR was performed using BM-96F and BM-1268R primer pairs specifically detecting CGMMV infection, whereas another primer set (BM-1180F and BM-1269R) was used for the specific detection of the DG (PDS)-1 replicon. The replication pattern of DG (PDS)-1 was detected in the 0 to 9 DPI period. Here, lanes 1–11 represent the samples (leaf discs) harvested at 0 DPI, 1 DPI, 2 DPI, 3 DPI, 4 DPI, 5 DPI, 6 DPI, 7 DPI, 8 DPI, 9 DPI, and 9 DPI from the 10 specified leaves of the 10 infiltrated plants. Lane 11 is the only symptomatic (BP4) CGMMV-infected plant. Lane 12 is the healthy plant; whereas, lanes 13 and 14 are the BP4 and DG (PDS)-1 plasmid controls, with a reagent (negative) control in lane 15. (**b**) The DG(PDS)-1 replicon was detected from 2–3 DPI onwards in the first systemic leaves that were tested 1–10DPI, whereas it is absent in a helper-type CGMMV-infected plant (BP4, BP7) and health plant (H). (**c**) Similarly, the DG(PDS)-1 replicon was detected in the second systemic leaf from 3 DPI onwards and tested for up to 8 DPI. In the whole study, DG(PDS)-1 and DG-1 were used as the plasmid control and (-) ve was the reagent control.

**Figure 7 plants-13-01414-f007:**
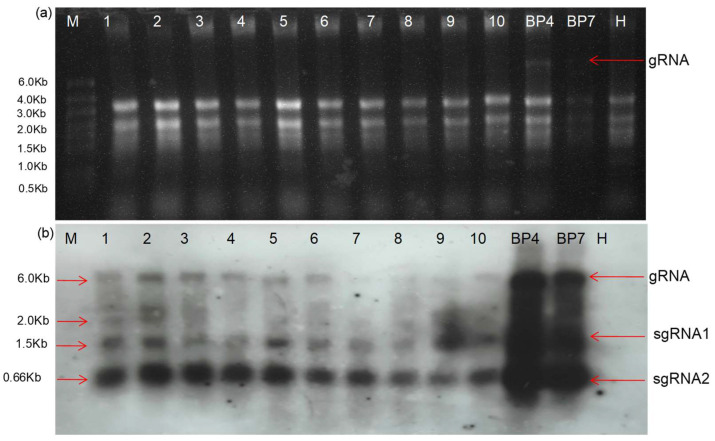
An accumulation of the DG-1 replicon (2.0 kb) was evident from the Northern blot assay. (**a**) The separation of RNA in the denaturing gel. The RNA was extracted from the first to tenth leaves (as numbered) of CGMMV-infected *N. benthamaina* infiltrated with pDG(PDS)-1. BP4 and BP7 represent symptomatic and asymptomatic CGMMV-infected *N.benthamiana* without DG(PDS)-1 infiltration, while the RNA of the healthy plant sample (H) is used as the control. The gRNA and sgRNA denote the genomic RNA and subgenomic RNA, respectively. (**b**) Pictorial depiction of Northern blot indicating presence of DG(PDS)-1 in first and second systemic leaves (marked by 2.0 Kb).

**Figure 8 plants-13-01414-f008:**
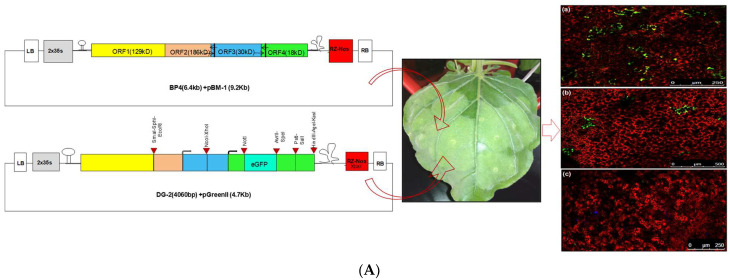
(**A**) The restricted localisation of DG-2 within the cellular system of leaf lamina, showing the scattered expression of GFP in yellowish-green (**a**,**b**), whereas it was absent in the healthy tissue, showing as a red colour (**c**). The expression was visualised in confocal laser microscopy. The DG-2 construct was agroinfiltrated into CGMMV-infected *N. benthamiana*, 7 days post-infiltration of the CGMMV infectious clone (BP4). (**B**) In planta cellular expression of GFP by DG-2 construct of CGMMV at different temporal scales when agroinfiltrated into CGMMV-infected *N. benthamiana* plant. Expression of GFP 2 days post-infiltration (DPI) of DG-2 (**a**) and 4 DPI (**b**), 6 DPI (**c**), 8 DPI (**d**), 10 DPI (**e**), and 12 DPI (**f**) was examined using confocal microscopy. It indicates that the critical time for the higher expression of GPF, as well as other foreign proteins, by this construct lies within the 6–10 DPI span. Here, the expression of GFP was not visualised in the only-CGMMV-infected plant (**g**) and the only-DG2-infiltrated plant tissue (**h**).

**Figure 9 plants-13-01414-f009:**
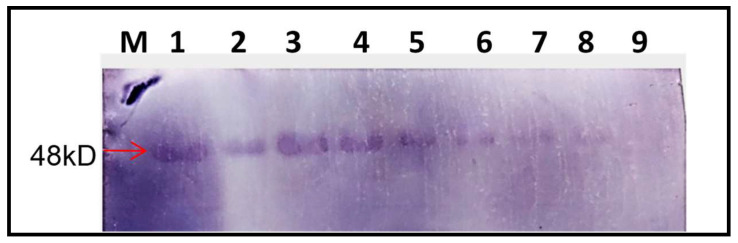
The Western blot assay of the crude plant protein showed the expression of the GFP and CP-fused protein (48 kDa) of CGMMV using a polyclonal anti-GFP antibody. The change in the expression level was visualised on different timescales. The expression of GFP increases from the 2nd DPI (1) to the 4th DPI (2) and then the 6th DPI (3); after that, it gradually decreases from the 8th DPI (4) onwards but remains at a detectable level up to the 10th DPI (5), 12th DPI (6), and 15th DPI. (7) The GFP expression level can be compared with pCambia1302 (8). Here, M indicates a protein marker and 9 represents the crude protein from a healthy plant.

**Figure 10 plants-13-01414-f010:**
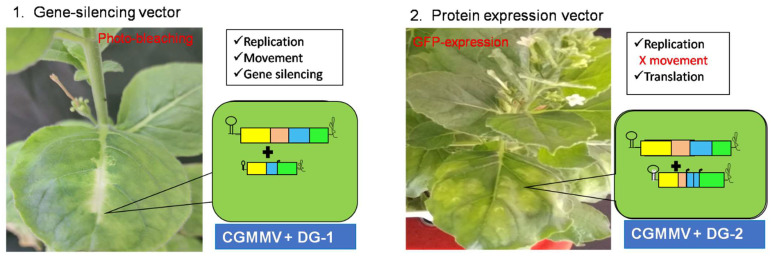
Development of a bipartite vector system using CGMMV genome. The trans-replication and systemic movement of the deconstructed genome, DG1, make it suitable for use as a gene-silencing vector; whereas the replication and translational ability of another deconstructed genome, DG2, ensure its promising use as a protein expression vector.

## Data Availability

Data are contained within the article and Appendix A.

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
