# Peer review of "Expanding Possibilities for Foreign Gene Expression by Cucumber Green Mottle Mosaic Virus Genome-Based Bipartite Vector System"

_plants, 2024, doi:10.3390/plants13101414_

Round 1
Reviewer 1 Report
Comments and Suggestions for Authors
Authors Anirudha Chattopadhyay and coworkers presented here a manuscript entitled “Expanding Possibilities for Foreign Gene Expression by CGMMV Genome-Based Bipartite Vector System”
The authors describe the successful modification of cucumber green mottle mosaic virus (CGMMV) for use in expressing foreign genes in cucurbit plants.
They developed two modified constructs based on the CGMMV genome mimicking natural subgenomic viral RNAs or satellite RNAs to increase the expression level of inserted genes and the possibility of inserting more genes or even longer genes.
The modified CGMMV genome was used in the model plant Nicotiana benthamiana and with the model gene gfp.
Their work is an important contribution to understanding the role of subgenomic viral elements and, from a practical point of view, also an important tool for functional genomics of plant viruses and expression of foreign genes in plants.
The following changes are needed to improve the clarity and readability of the manuscript:
The description of the construction of the edited versions of the CGMMV genome in the materials should be clarified with a graphical diagram. Therefore, I recommend moving the relevant figures from the supplements to the main body of the manuscript.
Sequences of primers used in the work are missing.
Avoid mentioning unpublished results (lines 118, 164). Instead, briefly introduce the materials or techniques used.
Line 235: mentioned Table S1 is not present in the Supplements.
Line 652: Figure 7 lacks description of individual lanes.
Line 716: subgenomic virus particles – what is that?
Line 738: subviral particles – what is that?
The paper is worthy of publication in Plants after minor revision.
Author Response
Reviewer 1:
Authors Anirudha Chattopadhyay and coworkers presented here a manuscript entitled “Expanding Possibilities for Foreign Gene Expression by CGMMV Genome-Based Bipartite Vector System”. The authors describe the successful modification of cucumber green mottle mosaic virus (CGMMV) for use in expressing foreign genes in cucurbit plants. They developed two modified constructs based on the CGMMV genome mimicking natural subgenomic viral RNAs or satellite RNAs to increase the expression level of inserted genes and the possibility of inserting more genes or even longer genes. The modified CGMMV genome was used in the model plant Nicotiana benthamiana and with the model gene gfp. Their work is an important contribution to understanding the role of subgenomic viral elements and, from a practical point of view, also an important tool for functional genomics of plant viruses and expression of foreign genes in plants.
The following changes are needed to improve the clarity and readability of the manuscript:
- The description of the construction of the edited versions of the CGMMV genome in the materials should be clarified with a graphical diagram. Therefore, I recommend moving the relevant figures from the supplements to the main body of the manuscript.
Authors reply
Thank you very much, sir, figures S1 and S4 are moved from the supplementary materials to the main text to present the constructs designing an understandable manner.
- Sequences of primers used in the work are missing
Authors reply
Thank you very much sir for addressing this issue. All the primer sequences used in this study were tabulated in supplementary figure no: Table S1 and are incorporated in the supplementary file.
- Avoid mentioning unpublished results (lines 118, 164). Instead, briefly introduce the materials or techniques used.
Authors reply
Sorry sir, a brief introduction of the unpublished result may create problems in publishing the result further. Previously, we already faced such a problem.
- Line 235: mentioned Table S1 is not present in the Supplements.
Authors reply
Sorry sir, we missed Table S1 in the supplementary file, and now it is being included.
- Line 652: Figure 7 lacks a description of individual lanes
Authors reply
Thanking you sir for your suggestion. The description is incorporated within Figure 7 to represent GFP expression at different temporal scales.
6.Line 716: subgenomic virus particles – what is that?
Authors reply
Subgenomic RNAs are generated by various positive-strand viruses as a result of imperfect replication processes, allowing for strategic expression of specific proteins or regulation of their viral life cycle. These RNAs typically possess identical 3' ends to the genomic RNA but exhibit deletions at the 5' ends, aligning the 5' end of the RNA with the start codon of downstream open reading frames (ORFs) on the genomic RNA.
7.Line 738: subviral particles – what is that?
Authors reply
Subviral particles are infectious agents that resemble viruses in structure and composition, notably smaller and simpler than viruses but sometimes lack some key components like protein capsid. They mostly depend on a specific helper virus for their replication, movement and transmission.
Reviewer 2 Report
Comments and Suggestions for Authors
In this paper, the authors describe the development of a bipartite vector system utilizing defective RNA (dRNA) derived from CGMMV. Two dRNA vectors (DG-1 and DG-2) were designed and synthesized. The authors reported that DG-1 induced systemic VIGS of PDS following additional inoculation of wild-type CGMMV infected N. benthamiana, while DG-2 expressed GFP in agro-infiltrated tissues but did not cause systemic infection. However, previous study has already demonstrated the production of useful proteins using bipartite vector systems with tobamovirus. Additionally, despite utilizing CGMV in this study, effective results were only observed in N. benthamiana. Although the artificial design of DG-2 is intriguing, a fully systemic infectious bipartite vector system has not been established yet. Furthermore, the potential advantages of the bipartite system, such as the stability of large size inserts, have not been thoroughly analyzed. Consequently, the novelty and impact of this study appear to be limited.
Moreover, the organization of the paper and the presentation of figures and tables require substantial improvement. Major revisions of the text and figures are necessary. For instance, detailed descriptions of construct generation and the process of establishing the virus detection method may be omitted from the results section. Additionally, clarity is lacking in the interpretation of electrophoresis images as it is not indicated which band represents the viral genome. Furthermore, for virus infection analysis, Northern blotting should be utilized instead of RT-PCR to avoid potential amplification of plasmid DNA at the infiltration site.
Author Response
Reviewer 2:
In this paper, the authors describe the development of a bipartite vector system utilizing defective RNA (dRNA) derived from CGMMV. Two dRNA vectors (DG-1 and DG-2) were designed and synthesized. The authors reported that DG-1 induced systemic VIGS of PDS following additional inoculation of wild-type CGMMV-infected N. benthamiana, while DG-2 expressed GFP in agro-infiltrated tissues but did not cause systemic infection. However, previous study have already demonstrated the production of useful proteins using bipartite vector systems with tobamovirus. Additionally, despite utilizing CGMV in this study, effective results were only observed in N. benthamiana. Although the artificial design of DG-2 is intriguing, a fully systemic infectious bipartite vector system has not been established yet. Furthermore, the potential advantages of the bipartite system, such as the stability of large size inserts, have not been thoroughly analyzed. Consequently, the novelty and impact of this study appear to be limited.
- Moreover, the organization of the paper and the presentation of figures and tables require substantial improvement. Major revisions of the text and figures are necessary. For instance, detailed descriptions of construct generation and the process of establishing the virus detection method may be omitted from the results section.
Authors reply
Improving the organization of the paper and enhancing the presentation of figures and tables could be beneficial. However, the descriptions of construct design and their generation are integral parts of the manuscript and cannot be omitted. Additionally, detailing the process of establishing the virus detection method is necessary for a comprehensive presentation of results.
- Additionally, clarity is lacking in the interpretation of electrophoresis images as it is not indicated which band represents the viral genome.
Authors reply
In the case of tobamoviruses (including CGMMV), simultaneous in vivo/in planta replication of multiple genomic strands (in the form of replicating fork) is evident. Consequently, amplification of multiple incomplete strands using terminal/UTR primers is commonly observed in gel electrophoresis, making it challenging to interpret all these amplifications. Furthermore, the CGMMV complete genome size is approximately 6.4 kb, which presents difficulty in amplification using standard Taq polymerase enzyme with an extension time of 30 seconds at 72°C.
- Furthermore, for virus infection analysis, Northern blotting should be utilized instead of RT-PCR to avoid potential amplification of plasmid DNA at the infiltration site.
Authors reply
Thank you for your insightful suggestion regarding the virus infection analysis. We acknowledge the potential issue of plasmid DNA amplification at the infiltration site when using RT-PCR and agree that Northern blotting could offer a more accurate alternative. However, it's worth noting that we utilized Total RNA as a negative control without cDNA synthesis in the PCR reaction. This approach allows us to detect any potential plasmid contamination, as the negative control of total RNA would provide amplification in PCR if such contamination were present. We observed no amplification in the RNA used as a control, indicating the absence of plasmid contamination.
Reviewer 3 Report
Comments and Suggestions for Authors
The paper discusses replication-deficient expression vectors that are based on the genome of cucumber green mottle mosaic virus (CGMMV). The authors demonstrate that these deconstructed genomes can replicate and move systemically in CGMMV-infected plants. However, an attempt to use one of the reported constructs for silencing the PDS gene was not very successful. Additionally, the expression of GFP from another version of the deconstructed vector resulted in a patchy expression phenotype. In my opinion, the paper is suitable for publication in Plants. However, I have concerns that should be addressed before the paper is accepted for publication.
Major points
1. The main paper text should include a schematic representation of the reported deconstructed genomes. Therefore, Figures S1 and S4 should be moved from the Supplementary Materials to the main text. Additionally, these figures should be revised to present the constructs in a more realistic and understandable manner. The figures in their current form may be challenging to interpret. Additionally, it may be beneficial to exclude unnecessary details, such as restriction sites and elements in the UTRs. It is recommended to depict the genomes to scale.
2. Lines 557-559. "These results provided phenotypic evidence of the functional significance of the pDG-1 construct as a successful VIGS vector."
The statement that the construct is a successful VIGS vector is false. The data presented in Figure S10 indicate that the pDG-1 construct is inefficient in inducing systemic gene silencing in plants, as only small patches exhibiting silencing were observed on a few leaves. Section 3.3.1.2 should be revised accordingly.
3. The level of GFP expressed from a deconstructed vector should be quantified and compared to other known expression systems, for example 35S promoter-driven expression or expression from a non-deconstructed tobamovirus genome such as the 30B construct (Shivprasad et al.,Virology 255, 312-323,1999).
Minor points
The last sentence of the abstract should be removed, as the abstract should not contain self-promoting language and should describe the reported results.
What is dRNA? The text does not provide an explanation for this abbreviation, which is not commonly known.
A link between predicted free energy of RNA secondary structure (lines 339-346) and the genome stability is not self-evident and should be explained.
Author Response
Reviewer 3:
Comments and Suggestions for Authors
The paper discusses replication-deficient expression vectors that are based on the genome of cucumber green mottle mosaic virus (CGMMV). The authors demonstrate that these deconstructed genomes can replicate and move systemically in CGMMV-infected plants. However, an attempt to use one of the reported constructs for silencing the PDS gene was not very successful. Additionally, the expression of GFP from another version of the deconstructed vector resulted in a patchy expression phenotype. In my opinion, the paper is suitable for publication in Plants. However, I have concerns that should be addressed before the paper is accepted for publication.
Major points
- The main paper text should include a schematic representation of the reported deconstructed genomes. Therefore, Figures S1 and S4 should be moved from the Supplementary Materials to the main text. Additionally, these figures should be revised to present the constructs in a more realistic and understandable manner. The figures in their current form may be challenging to interpret. Additionally, it may be beneficial to exclude unnecessary details, such as restriction sites and elements in the UTRs. It is recommended to depict the genomes to scale.
Authors reply
Thank you for your observation, sir. We have relocated the schematic representation of the reported deconstructed genomes from the supplementary materials to the main text to illustrate the construct design more prominently. Additionally, as the genomes were initially presented at their full scale, we believe that including the restriction sites (intentionally positioned for multiple MCS, particularly in Figure S4) and UTR elements (crucial regulatory components for replication) is crucial for understanding the constructs' intended purpose. This ensures clarity for both molecular biologists and virologists reviewing the manuscript.
- Lines 557-559. "These results provided phenotypic evidence of the functional significance of the pDG-1 construct as a successful VIGS vector."
The statement that the construct is a successful VIGS vector is false. The data presented in Figure S10 indicate that the pDG-1 construct is inefficient in inducing systemic gene silencing in plants, as only small patches exhibiting silencing were observed on a few leaves. Section 3.3.1.2 should be revised accordingly.
Authors reply
Apologies, sir. We believe these phenotypic pieces of evidence are significant. Previously, Liu et al. (2020) reported a slight photobleaching effect on Nicotiana benthamiana leaves using a CGMMV full-length genome-based VIGS vector expressing various lengths of the NbPDS gene (Doi:10.1186/s13007-020-0560-3). However, it's important to note that Nicotiana benthamiana is an experimental host for CGMMV, not a natural host. Therefore, phenotypic evidence may be restricted to limited areas.
- The level of GFP expressed from a deconstructed vector should be quantified and compared to other known expression systems, for example 35S promoter-driven expression or expression from a non-deconstructed tobamovirus genome such as the 30B construct (Shivprasad et al.,Virology 255, 312-323,1999).
Authors reply
The level of GFP expressed from a deconstructed vector has already been quantified in Supplementary Table S5. Additionally, the TMV genome-based 30B vector is hybrid, and its experimental setup was substantially different. Therefore, direct comparison between the two vectors would be challenging unless they were evaluated under the same experimental conditions.
- The last sentence of the abstract should be removed, as the abstract should not contain self-promoting language and should describe the reported results.
Authors reply
Thank you very much sir, we removed these lines.
- What is dRNA? The text does not provide an explanation for this abbreviation, which is not commonly known.
Authors reply
Thank you sir for your observation, Here, deconstructed RNA of the CGMMV genome was referred as dRNA.
- A link between predicted free energy of RNA secondary structure (lines 339-346) and the genome stability is not self-evident and should be explained
Authors reply
Thank you for your input. The genome stability of RNA viruses is typically associated with the predicted free energy of RNA secondary structure. RNA viruses depend on stable secondary structures for various processes such as replication, transcription, and interaction with host factors. Alterations in RNA secondary structure stability often impact the replication fidelity of RNA viruses. Lower values of ΔG indicate more stable secondary structures; therefore, deconstructed RNA molecules with lower ΔG values are less likely to undergo structural changes.
Round 2
Reviewer 2 Report
Comments and Suggestions for Authors
This revised version likely deserves to be published.
Author Response
Dear Reviewer, thank you for your review and feedback regarding our revised MS.
Thank you
Abdul
Reviewer 3 Report
Comments and Suggestions for Authors
The corrections made have improved the manuscript. However, there are still two issues that need to be resolved before the paper can be accepted for publication.
The statement that the genome stability of RNA viruses is typically associated with the predicted high free energy of RNA secondary structure is unjustified. In fact, such a link is not described in the literature. If the authors insist on making this statement, references to the relevant publications must be provided in the text of the paper. In addition, the reason for calculating the free energy should be clearly described in the text.
Line 584. I do not agree with the authors' argumentation regarding the efficiency of the reported VIGS vector. At the very least, the word "successful" should be removed from the sentence "...significance of the pDG-1 construct as a successful VIGS vector".
Author Response
Dear Reviewer, Thank you for your time and efforts, please see our response below.
1. Comments to Author: The statement that the genome stability of RNA viruses is typically associated with the predicted high free energy of RNA secondary structure is unjustified. In fact, such a link is not described in the literature. If the authors insist on making this statement, references to the relevant publications must be provided in the text of the paper. In addition, the reason for calculating the free energy should be clearly described in the text.
Reply: Apologies, In RNA biology, the stability of an RNA genome is determined by its secondary structures, characterized by their configuration and minimum free energy values. There is ample evidence in the literature supporting this notion.
‘RNA structure prediction techniques based on free energy minimization are typically used on a single RNA sequence [25]. Energetically, the most stable RNA secondary structure(s) of a plant virus genome contains canonical A:U, G:U, and G:C base pairs to arrange the structure into a conventional helical form. Thus, it is considered that the lower the energy, the more stable the structure is [26]’
References added in text
- Mathews DH, Turner DH. Prediction of RNA secondary structure by free energy minimization. Curr Opin Struct Biol. 2006 Jun;16(3):270-8. doi: 10.1016/j.sbi.2006.05.010.
- Zuker M, Stiegler P. Optimal computer folding of large RNA sequences using thermodynamics and auxiliary information. Nucleic Acids Res. 1981;9:133–48.
2. Comments to Authors: Line 584. I do not agree with the authors' argument regarding the efficiency of the reported VIGS vector. At the very least, the word "successful" should be removed from the sentence "... the significance of the pDG-1 construct as a successful VIGS vector".
Reply.. agreed and "successful" was removed and the sentence is rewritten as “..phenotypic evidence of the functional significance of the pDG-1 construct as a VIGS vector”.